# Confusion-Resistant Federated Learning via Diffusion-Based Data Harmonization on Non-IID Data

**Xiaohong Chen**[1,2,3]    **Canran Xiao**[1]*   **Yongmei Liu**[1,4]

[1] School of Business, Central South University, Changsha, Hunan 410083, China
[2] Xiangjiang Laboratory, Changsha, Hunan 410205, China
[3] School of Advanced Interdisciplinary Studies, School of Management Science and Engineering, Hunan University of Technology and Business, Changsha, Hunan 410205, China
[4] Urban Smart Governance Laboratory, Changsha, Hunan 410083, China
`c88877803@163.com, xiaocanran@csu.edu.cn, liuyongmeicn@163.com`

## Abstract

Federated learning has become a pivotal distributed learning paradigm, involving collaborative model updates across multiple nodes with private data. However, handling non-i.i.d. (not identically and independently distributed) data and ensuring model consistency across heterogeneous environments present significant challenges. These challenges often lead to model performance degradation and increased difficulty in achieving effective communication among participant models. In this work, we propose Confusion-Resistant Federated Learning via Consistent Diffusion (CRFed), a novel framework designed to address these issues. Our approach introduces a new diffusion-based data harmonization mechanism that includes data augmentation, noise injection, and iterative denoising to ensure consistent model updates across non-i.i.d. data distributions. This mechanism aims to reduce data distribution disparities among participating nodes, enhancing the coordination and consistency of model updates. Moreover, we design a confusion-resistant strategy leveraging an indicator function and adaptive learning rate adjustment to mitigate the adverse effects of data heterogeneity and model inconsistency. Specifically, we calculate importance sampling weights based on the optimal sampling probability, which guides the selection of clients and the sampling of their data, ensuring that model updates are robust and aligned across different nodes. Extensive experiments on benchmark datasets, including MNIST, FashionMNIST, CIFAR-10, CIFAR-100, and NIPD, demonstrate the effectiveness of CRFed in improving accuracy, convergence speed, and overall robustness in federated learning scenarios with severe data heterogeneity.

## 1   Introduction

Federated Learning (FL) [McMahan et al., 2017b] has emerged as a powerful paradigm for distributed machine learning, enabling multiple clients to collaboratively train a shared model without exchanging raw data. This approach addresses critical concerns around data privacy and security, which are increasingly significant in various sectors such as healthcare [Antunes et al., 2022], finance [Chatterjee et al., 2023], and IoT [Li et al., 2020a, Pan et al., 2023, Yao et al., 2024]. However, one of the fundamental challenges in FL is dealing with non-independent and identically distributed (non-IID) data, which can significantly impair the performance and convergence of the global model [Zhu et al., 2021].

---

*Corresponding author

38th Conference on Neural Information Processing Systems (NeurIPS 2024).

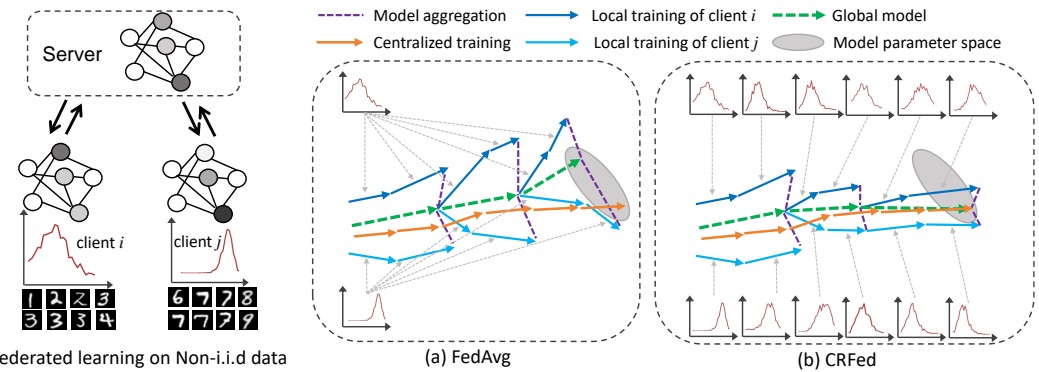

Figure 1: Problem illustration of federated learning on Non-i.i.d data.

As illustrated in Figure 1, FL on non-IID data often suffers from issues like divergent model updates and inconsistent global models. Client models trained on heterogeneous data distributions tend to diverge [Ye et al., 2023], making it difficult for the server to aggregate them into a coherent global model. This divergence is due to inconsistencies in data sources [Xiao and Liu, 2024] and distributions across clients [Duan et al., 2021]. This problem leads to reduced accuracy and slower convergence rates, highlighting the need for effective solutions to handle data heterogeneity. Existing research has made significant strides in improving the robustness and efficiency of FL in non-IID settings. Notable methods include FedProx [Li et al., 2020b], which adds a proximal term to handle heterogeneity. Techniques such as MOON [Li et al., 2021b]and FedGen [Nguyen et al., 2021] introduce sophisticated strategies like contrastive learning and data generation to mitigate the effects of data heterogeneity. Despite these advancements, issues related to data distribution disparities and model inconsistency persist, limiting the scalability and effectiveness of FL in real-world scenarios.

Given the critical gaps in existing FL approaches, particularly their limited robustness to severe non-IID data distributions, there is a pressing need for more resilient and adaptive solutions. This research is motivated by the necessity to enhance FL's capability to handle heterogeneous data efficiently. The primary objective of this study is to develop a novel FL framework, Confusion-Resistant Federated Learning via Consistent Diffusion (CRFed), which integrates advanced mechanisms to address data distribution disparities and enhance model consistency across clients.

Our contributions can be summarized as follows:

1. We propose a novel indicator function that dynamically adjusts sample weighting based on loss values and uncertainties, facilitating a self-paced learning approach that prioritizes more difficult samples over time.

2. Our framework employs a diffusion-based mechanism to harmonize data distributions, involving iterative noise injection and denoising processes that align local data with the desired distribution.

3. We implement a strategic client selection method based on the indicator function, ensuring the inclusion of the most reliable clients, which enhances the robustness and consistency of model updates.

4. Our extensive experimental evaluations demonstrate that CRFed achieves state-of-the-art performance on benchmark datasets. CRFed outperforms existing methods significantly in terms of accuracy and convergence speed under various non-IID settings.

## 2 Related Work

### 2.1 Non-IID Challenge in Federated Learning

The issue of non-IID data in FL was initially highlighted by FedAVG [McMahan et al., 2017a], and it has since been demonstrated that this challenge can significantly hinder the convergence and overall performance of the global model [Zhao et al., 2018, Li et al., 2019]. Numerous studies, categorized as client-centric methods, have been proposed to tackle this problem by adjusting the local training

objectives using insights from the global model and the local models of other clients [Wang et al., 2021]. For instance, FedProx [Li et al., 2020b] introduced a proximal term to constrain local updates by leveraging the global model. SCAFFOLD [Karimireddy et al., 2020] utilized control variates to correct for local training drift, while FedDyn [Acar et al., 2021] introduced a dynamic regularizer for parallelizing gradients among clients. MOON [Li et al., 2021b] applied contrastive learning to minimize the discrepancy between model representations, thereby correcting local training.

Despite their contributions, these methods fall short of fully resolving the core of the non-IID issue and may experience performance limitations in scenarios with highly skewed data distributions [Li et al., 2022]. Beyond client-side adjustments, the server also plays a role in mitigating the adverse effects of non-IID data by calibrating the biased global model post-aggregation. For example, CCVR [Luo et al., 2021] uses virtual representations from an approximated Gaussian mixture model to correct the classifier. FedFTG [Zhang et al., 2022] employs data-free knowledge distillation to refine the global model with the knowledge derived from local models. Additionally, strategies such as client clustering [Ghosh et al., 2020, Long et al., 2023] and client selection [Zhang et al., 2021, Wang et al., 2020], can be implemented by the server to alleviate the non-IID problem. IFCA [Ghosh et al., 2020] iteratively estimates client cluster identities based on local empirical loss and updates model parameters for each cluster via gradient descent.

## 2.2 Importance Sampling in Federated Learning

In federated learning (FL), data sampling strategies are vital for enhancing distributed training efficiency. [Tuor et al., 2020] proposed selecting local training data based on user-end data correlation analysis. This led to dynamic sampling strategies like [Li et al., 2021a], where training sample importance is determined by model gradient magnitudes. Similarly, [Rizk et al., 2022] used gradient norms to derive sampling weights, minimizing theoretical convergence bounds. However, these methods require immediate gradient computations, increasing local overhead, and assume convex loss functions, which may not apply to deep learning models [Rizk et al., 2021]. Therefore, developing importance sampling methods suitable for deep learning-based FL remains an open challenge.

FL convergence can be theoretically analyzed due to the model aggregation mechanism [98, 101-102, 150], with experimental validation for deep learning tasks [Wan et al., 2021]. Most studies rely on theoretical derivations, limiting practical application. This study aims to use a diffusion model to automate the modeling of optimal sampling strategies in FL.

# 3 Method

## 3.1 Overview

The CRFed framework, shown in Figure 2, addresses challenges posed by non-i.i.d. data in FL. Our approach integrates a diffusion mechanism and a confusion-resistant strategy to ensure consistent and robust model updates across heterogeneous data distributions. The core idea of CRFed is that the performance of client $i$'s data on the global model reflects its contribution to the training process. By using an optimal indicator function, we determine the optimal data sampling probability for each client, enhancing training efficiency and model performance.

The framework comprises several key components: the current global model downloaded by clients at time $t$; the Model Encoder and Meta-model, which process the global model and client-specific data for the diffusion process; an indicator function, computed using client $i$'s data on the global model; the Diffusion-based Data Harmonization Mechanism, which uses data augmentation, noise injection, and probabilistic modeling to mitigate data distribution disparities; and the Distribution Decoder, which aligns the denoised data distribution with the desired distribution.

## 3.2 Indicator Function and Meta-model

The Indicator Function $I_\lambda(l_i, \sigma_i)$ is designed to measure the reliability of the $i$-th sample's loss value $l_i$ and its associated uncertainty $\sigma_i$. The design is inspired by self-paced learning [Fan et al., 2017, Castells et al., 2020], which adjusts the weights of samples based on their loss values, allowing for a gradual learning process from easy to difficult samples. In the context of federated learning, this means that each client should adopt a self-paced learning paradigm, sampling its data in a way that allows the global model to learn from simple to complex tasks. The Indicator Function captures this performance and is defined as follows:

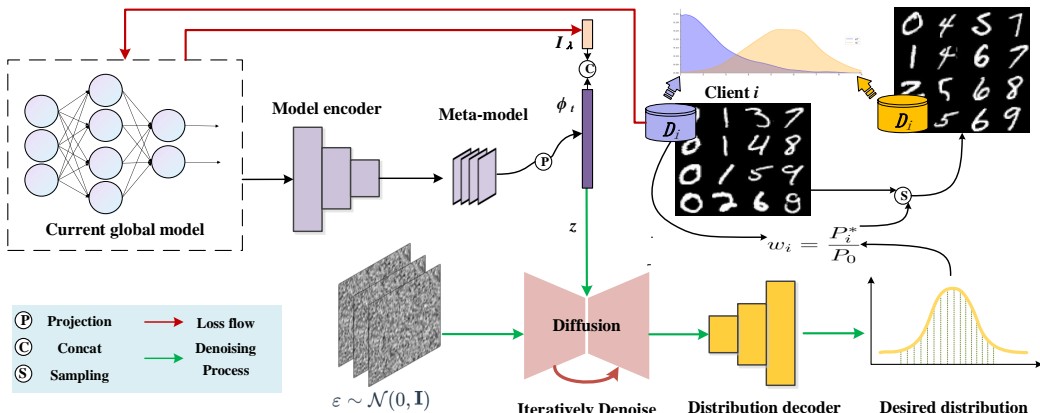

Figure 2: **CRFed Framework.** The process begins with the current global model, which is downloaded by clients. The model encoder processes the global model, and the meta-model is obtained. This meta-model is then projected into a higher-dimensional space and concatenated with the indicator function, forming the combined representation $z_i$. The diffusion-based data harmonization mechanism adds noise to this representation and iteratively denoises it to achieve the desired distribution. The distribution decoder then aligns the denoised data distribution. Client $i$'s data is sampled based on importance sampling weights $w_i$, calculated as the ratio of the optimal sampling probability $P_i^*$ to the original data distribution $P_0$. This ensures that the sampled data aligns with the desired distribution, following a curriculum learning approach that progresses from easy to difficult samples, thus enhancing overall model performance.

$$I_\lambda(l_i, \sigma_i) = (l_i - \tau)\sigma_i + \lambda(\log \sigma_i)^2 \tag{1}$$

where $\lambda$ is a pre-set regularization coefficient. $\tau$ is a confidence threshold that determines the difficulty of the sample based on its loss value. It can either be a fixed constant or a dynamically adjusted weighted average during the training process.

The Indicator Function can be further explained using the following steps: for each client $i$, the loss value $l_i$ of each sample is calculated on the current global model, and the uncertainty $\sigma_i$ of each sample is estimated based on its difficulty. The Indicator Function $I_\lambda(l_i, \sigma_i)$ is then used to assign weights to the samples, with easier samples having lower weights and more difficult samples having higher weights. This adaptive weighting mechanism ensures that the model focuses more on difficult samples over time, leading to improved learning efficiency and robustness.

**Theorem 3.1.** *In the CRFed framework, using the indicator function $I_\lambda(l_i, \sigma_i)$ ensures stable and convergent updates for heterogeneous federated learning. For an appropriately chosen learning rate $\eta$, the model update rule for client $i$ at iteration $t$,*

$$\theta_{t+1} = \theta_t - \eta \left( \sigma_i^* + (l_i - \tau)k + 2\lambda \frac{\log \sigma_i^*}{\sigma_i^*} k \right) \nabla_\theta l_i, \tag{2}$$

*guarantees a decreasing step size, promoting convergence. Moreover, CRFed achieves a tighter bound on update steps than FedAvg, indicating faster convergence under the same conditions.*

The proof of Theorem 3.1 can be found in Appendix A.2.

Given the global model $\theta$, the optimal uncertainty $\sigma_i^*$ can be derived through the following theorem:

**Theorem 3.2.** *The optimal uncertainty $\sigma_i^*$ for a given loss value $l_i$ is obtained by minimizing the Indicator Function $I_\lambda(l_i, \sigma_i)$. The solution is given by:*

$$\sigma_i^*(l_i) = \exp \left( -W \left( \frac{1}{2\lambda} \max \left( -\frac{2}{e}, l_i - \tau \right) \right) \right) \tag{3}$$

*where $W(\cdot)$ is the Lambert W function.*

We rewrite the indicator function by setting $\sigma_i = e^{x_i}$, reduce the derivative condition to a Lambert $W$ form under domain constraints, and thus obtain a closed-form solution for $\sigma_i^*$; the complete derivation is presented in Appendix A.3.

For each sample indexed by $i$, we determine the optimal uncertainty $\sigma_i^*$ as a function of its loss $l_i$. Specifically, let $I_\lambda(l_i, \sigma_i)$ be the indicator function defined before. Then, selecting $\sigma_i^*$ that minimizes $I_\lambda(l_i, \sigma_i)$ induces an adaptive weighting scheme:

$$\omega_i(l_i) \; \propto \; \frac{1}{\sigma_i^*(l_i)}, \tag{4}$$

ensuring that when $l_i$ is relatively large, the corresponding optimal $\sigma_i^*$ decreases, thus increasing $\omega_i(l_i)$ and emphasizing more difficult samples. Conversely, smaller $l_i$ values lead to larger $\sigma_i^*$ and lower $\omega_i(l_i)$, indicating that easier samples receive diminished focus. As a result, the distribution defined by $\sigma_i^*$ is optimally aligned with the current global model $\theta_t$, conforming to the self-paced learning principle.

Since $I_\lambda(l_i, \sigma_i)$ can be regarded as quantifying each client's contribution relative to $\theta_t$, we embed $\theta_t$ within the diffusion model as follows: using an autoregressive encoder $E$, we compress $\theta_t$ to a meta-model $\phi_t = E(\theta_t)$. Subsequently, $\phi_t$ is projected into a high-dimensional representation $P(\phi_t)$, and concatenated with $I_\lambda(l_i, \sigma_i)$ to form

$$z_i \; = \; \mathrm{concat}\big(P(\phi_t), I_\lambda(l_i, \sigma_i)\big). \tag{5}$$

We then feed $z_i$ into the diffusion model, denoted by $\mathrm{DiffusionModel}(z_i)$, to refine the data distribution with respect to the global context. Further implementation details about the encoder $E$ are provided in Appendix A.4.

### 3.3 Diffusion-based Data Harmonization

The diffusion-based data harmonization mechanism is a critical component of the CRFed framework, responsible for mitigating data distribution disparities and ensuring consistent model updates across heterogeneous environments. The harmonization process is shown in Figure 3, which involves adding noise to the data distribution and then iteratively denoising it to achieve the desired distribution. The workflow of this mechanism can be divided into two main processes: the forward diffusion process and the reverse denoising process.

**Forward Diffusion Process** Suppose that a training sample $\mathbf{x}_0$ is of a certain distribution, denoted as $q(\mathbf{x}_0)$. In the forward diffusion process, Gaussian noise with variance $\beta_t \in (0, 1)$ is added gradually to the sample $\mathbf{x}_0$ for $T$ steps, resulting in a latent sample $\mathbf{x}_T \sim \mathcal{N}(0, \mathbf{I})$. The process is defined as follows:

$$q(\mathbf{z}_{1:T}|z_i) = \prod_{t=1}^{T} q(\mathbf{z}_t|\mathbf{z}_{t-1}), \tag{6}$$

$$q(\mathbf{z}_t|\mathbf{z}_{t-1}) = \mathcal{N}(\mathbf{z}_t; \sqrt{1 - \beta_t}\mathbf{z}_{t-1}, \beta_t\mathbf{I}). \tag{7}$$

Using notations $\alpha_t = 1 - \beta_t$ and $\bar{\alpha}_t = \prod_{s=1}^{t} \alpha_s$, the sample $\mathbf{z}_t$ can be defined directly as:

$$\mathbf{z}_t = \sqrt{\bar{\alpha}_t}z_i + \sqrt{1 - \bar{\alpha}_t}\boldsymbol{\epsilon}, \quad \boldsymbol{\epsilon} \sim \mathcal{N}(0, \mathbf{I}). \tag{8}$$

**Reverse Denoising Process** The reverse denoising process aims to sample reversely from $\mathbf{z}_T$ through transition probabilities $q(\mathbf{z}_{t-1}|\mathbf{z}_t)$ for timesteps $T - 1$ through 1, yielding a sample drawn from $q(z_i)$. The transition $q(\mathbf{z}_{t-1}|\mathbf{z}_t)$ is a Gaussian distribution, tractable when conditioned on $z_i$:

$$q(\mathbf{z}_{t-1}|\mathbf{z}_t, z_i) = \mathcal{N}(\mathbf{z}_{t-1}; \tilde{\boldsymbol{\mu}}_t(\mathbf{z}_t, z_i), \tilde{\beta}_t\mathbf{I}), \tag{9}$$

where the mean $\tilde{\boldsymbol{\mu}}_t$ and variance $\tilde{\beta}_t$ are calculated as:

$$\tilde{\boldsymbol{\mu}}_t(\mathbf{z}_t, z_i) = \frac{1}{\sqrt{\alpha_t}} \left( \mathbf{z}_t - \frac{1-\alpha_t}{\sqrt{1-\bar{\alpha}_t}} \boldsymbol{\epsilon}_t \right), \tag{10}$$

$$\tilde{\beta}_t = \frac{1-\bar{\alpha}_{t-1}}{1-\bar{\alpha}_t} \beta_t. \tag{11}$$

The reverse transition probability $p_\theta(\mathbf{z}_{t-1}|\mathbf{z}_t)$ relies on the entire data distribution and is approximated through a neural network:

$$p_\theta(\mathbf{z}_{t-1}|\mathbf{z}_t) = \mathcal{N}(\mathbf{z}_{t-1}; \boldsymbol{\mu}_\theta(\mathbf{z}_t, t), \boldsymbol{\Sigma}_\theta(\mathbf{z}_t, t)), \tag{12}$$

The detailed derivations and computations for these processes can refer to A.1. In the context of federated learning, the reverse denoising process starts from the optimal indicator function $I^*$ obtained in the forward diffusion process. By progressively denoising, we obtain the optimal sampling probability for client $i$, ensuring that the final data distribution aligns with the desired distribution. This minimizes the impact of data heterogeneity and ensures robust model updates across all clients.

### 3.4 Confusion-Resistant Strategy

The confusion-resistant strategy is designed to address the challenges posed by data heterogeneity and model inconsistency in federated learning. It consists of three key components: client selection based on the indicator function, data sampling using the diffusion-based harmonization mechanism, and adaptive learning rate adjustment.

**Client Selection Strategy** To mitigate the adverse effects of data heterogeneity, we select clients based on their indicator function $I_\lambda(l_i, \sigma_i)$, which quantifies the reliability of their data. Clients with the lowest indicator values, reflecting higher data reliability, are chosen for training. This approach follows the curriculum learning paradigm, where lower values indicate better data:

$$\text{Selected Clients} = \{i | I_\lambda(l_i, \sigma_i) \leq \gamma\}, \tag{13}$$

where $\gamma$ is a dynamically adjusted threshold ensuring the selection of the most suitable clients.

**Data Sampling Strategy** For each selected client, the optimal sampling probability $P_i^*$ is determined through the reverse denoising process, starting from the optimal indicator function $I^*$. This ensures that the sampled data aligns with the desired distribution, enhancing the robustness of model updates. The importance sampling weight $w_i$ is calculated as follows:

$$w_i = \frac{P_i^*}{P_0}. \tag{14}$$

where $P_0$ is the original data distribution. Using $w_i$, we sample the local training data ($\mathcal{D}_i^{\text{sampled}} = \text{Sample}(\mathcal{D}_i, w_i)$). This sampling ensures the effective sampling probability aligns with $P_i^*$.

The distribution decoder, which is implemented as an autoencoder, is then used to decode the denoised data distribution. The autoencoder is trained to map the denoised samples back to the desired distribution, further ensuring that the data used for training is aligned with the ideal distribution. Refer to the A.4 for more details of distribution decoder.

**Adaptive Learning Rate Adjustment** The learning rate $\eta_i$ for each client is adjusted based on the indicator function value, enhancing the influence of more reliable data:

$$\eta_i = \eta_0 \cdot \frac{I_\lambda(l_i, \sigma_i)}{\max_j I_\lambda(l_j, \sigma_j)}, \tag{15}$$

where $\eta_0$ is the base learning rate.

The complete computational process(pseudocode) of CRFed is provided in the A.5.

# 4 Experiments

## 4.1 Experiment Setup

**Datasets** Our experiments are conducted on four widely used benchmark datasets: MNIST [LeCun et al., 1998], Fashion-MNIST [Xiao et al., 2017], CIFAR-10 [Krizhevsky et al., 2009], and CIFAR-100 [Krizhevsky et al., 2009]. To simulate Non-IID data scenarios, we utilize the Dirichlet distribution [Yurochkin et al., 2019] to generate non-IID partitions with varied concentration parameters, $\beta$. Smaller values of $\beta$ lead to more imbalanced data distributions among clients, thereby increasing levels of data heterogeneity. In our experiments, we set $\beta$ to 0.5 to reflect this imbalance. In all experiments, we simulate a federated learning environment with 10 edge nodes, i.e., $K = 10$. For the MNIST and FashionMNIST datasets, each node has 600 data samples. For the CIFAR-10 dataset, each node has 500 data samples. For CIFAR-100, the partitioning strategy remains the same, ensuring that each client's local data distribution varies significantly, simulating real-world federated learning scenarios. MNIST and FashionMNIST datasets consist of grayscale images of size $28 \times 28$ pixels, with 10 classes. CIFAR-10 and CIFAR-100 contain color images of size $32 \times 32$ pixels.

Additionally, we use the NIPD dataset [Yin et al., 2023], a benchmark specifically designed for federated learning in person detection tasks with Non-IID data. This dataset provides a real-world non-IID scenario to test the generalization of CRFed.

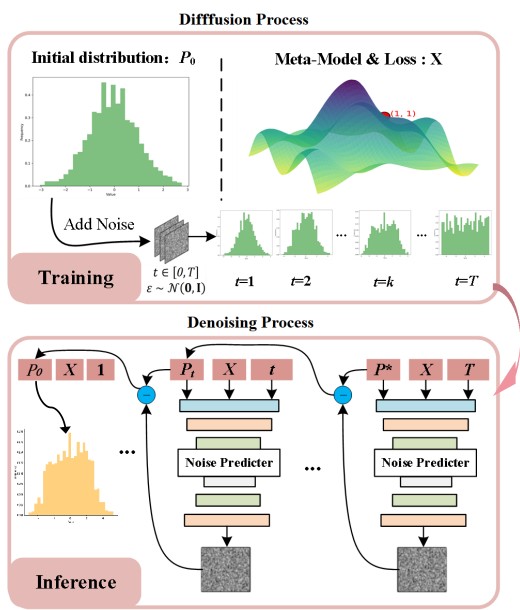

Figure 3: The diffusion-based data harmonization mechanism in CRFed framework. The process involves a forward diffusion process where Gaussian noise is added to the initial data distribution, transforming it into a latent representation. This is followed by a reverse denoising process that iteratively removes the noise, aligning the data distribution with the desired target distribution.

**Competing Methods** Apart from FedAvg [McMahan et al., 2017a], we compare the proposed algorithm with several benchmarking FL algorithms specialized for solving the non-IID problem, including FedProx [Li et al., 2020b], MOON [Li et al., 2021b], and FedGen [Nguyen et al., 2021]. We also compare our method against HFMDS-FL [Li et al., 2024], FRAug [Chen et al., 2023], G-FML [Yang et al., 2023], FedCD [Long et al., 2023], FedNP [Wu et al., 2023], and FedDPMS [Chen and Vikalo, 2023], which are recent state-of-the-art approaches addressing non-IID data issues in federated learning.

**Hyperparameter** For local training, the settings are as follows: MNIST with $E = 5$, $B = 10$, $\eta = 5 \times 10^{-3}$; FashionMNIST with $E = 5$, $B = 100$, $\eta = 2 \times 10^{-4}$; CIFAR-10 and CIFAR-100 with $E = 5$, $B = 100$, $\eta = 1 \times 10^{-4}$. Momentum optimization with a coefficient of 0.5 is applied. In the CRFed framework, key hyperparameters include maximum global rounds ($T_G$) set to 100, local training cycles ($E_l$) per global round set to 1, regularization coefficient ($\lambda$) set to 0.1, dynamically adjusted confidence threshold ($\tau$), and client selection threshold ($\gamma$) initially set to 0.5. These parameters are fine-tuned based on preliminary experiments to ensure training efficiency and model performance. The experiments were conducted using an NVIDIA GeForce RTX 4060 GPU, which has 8GB of VRAM. Detailed configurations of model structure are provided in A.6.

**Evaluation metrics** The primary evaluation metrics for our experiments focus on accuracy and the number of training rounds needed to reach convergence, addressing the challenges posed by non-IID data in federated learning. Accuracy is measured at the same training round across different models to ensure fair comparison. Convergence is assessed by the number of rounds required to achieve a target accuracy, which reflects the model's stability and efficiency. For the NIPD dataset,

we use mean Average Precision (mAP) as the evaluation metric. Following [Wang et al., 2020], all reported results are averaged over five runs with different random seeds to account for variability.

## 4.2 Performance Comparison

Table 1: Test accuracy of CRFed and the competing methods on five datasets. We run five trials with different random seeds and report the mean accuracy.

| Scheme | MNIST | FashionMNIST | CIFAR-10 | CIFAR-100 | NIPD (mAP) |
|---|---|---|---|---|---|
| FedAvg [McMahan et al., 2017a] | 0.976 | 0.847 | 0.650 | 0.362 | 0.821 |
| FedProx [Li et al., 2020b] | 0.978 | 0.844 | 0.655 | 0.365 | 0.826 |
| MOON [Li et al., 2021b] | 0.980 | 0.846 | 0.674 | 0.372 | 0.836 |
| FedGen [Nguyen et al., 2021] | 0.982 | 0.862 | 0.672 | 0.369 | 0.841 |
| HFMDS-FL [Li et al., 2024] | 0.982 | 0.868 | 0.678 | 0.377 | 0.846 |
| FRAug [Chen et al., 2023] | 0.981 | 0.865 | 0.675 | 0.374 | 0.851 |
| G-FML [Yang et al., 2023] | 0.983 | 0.870 | 0.681 | 0.378 | 0.854 |
| FedCD [Long et al., 2023] | 0.982 | 0.867 | 0.677 | 0.376 | 0.861 |
| FedNP [Wu et al., 2023] | 0.982 | 0.869 | 0.680 | 0.377 | 0.863 |
| FedDPMS [Chen and Vikalo, 2023] | 0.983 | 0.876 | 0.680 | 0.386 | 0.871 |
| CRFed | **0.985** | **0.878** | **0.683** | **0.389** | **0.882** |

**Accuracy comparison** Table 1 presents the test accuracy of CRFed compared to several federated learning algorithms under a highly heterogeneous setting ($\beta = 0.5$). Our proposed method shows notable improvements over FedAvg [McMahan et al., 2017a], with relative gains of 0.9% on MNIST, 3.7% on FashionMNIST, 5.1% on CIFAR-10, 7.5% on CIFAR-100, and a significant 7.4% improvement in mAP on the NIPD dataset. This highlights CRFed's robustness in handling non-IID data distributions. CRFed consistently outperforms all other methods across the datasets, underscoring its effectiveness in federated learning scenarios with severe data heterogeneity.

**Effect of Data Heterogeneity** We analyze the impact of data heterogeneity on the performance of the top 5 models by varying the Dirichlet concentration parameter $\beta$. Table 2 shows the performance of these models on CIFAR-100 and NIPD datasets for $\beta$ values ranging from 0.1 to 0.5. As expected, the performance generally decreases with smaller $\beta$ values due to increased data heterogeneity. Table 2 indicates that as $\beta$ decreases,

Table 2: Performance of top 5 models on CIFAR-100 and NIPD datasets under different $\beta$ values.

| | CIFAR-100 | | | NIPD (mAP) | | |
|---|---|---|---|---|---|---|
| Scheme | 0.1 | 0.3 | 0.5 | 0.1 | 0.3 | 0.5 |
| FedDPMS | 0.270 | 0.330 | 0.386 | 0.751 | 0.810 | 0.871 |
| FRAug | 0.268 | 0.328 | 0.374 | 0.746 | 0.800 | 0.851 |
| G-FML | 0.265 | 0.329 | 0.378 | 0.748 | 0.802 | 0.854 |
| FedCD | 0.275 | 0.333 | 0.376 | 0.750 | 0.808 | 0.861 |
| CRFed | **0.280** | **0.345** | **0.389** | **0.760** | **0.820** | **0.882** |

representing higher data heterogeneity, the performance of all models declines. CRFed consistently outperforms other methods across different $\beta$ settings, demonstrating its robustness in handling data heterogeneity. Notably, the relative performance gap between CRFed and other methods widens as $\beta$ decreases, highlighting its efficacy in more challenging federated learning scenarios.

**Effect of Increasing Edge Nodes** Figure 4 presents the performance of the top 5 models on CIFAR-100 and NIPD datasets as the number of edge nodes $K$ increases from 10 to 100. Across all models, performance generally improves with higher $K$ values, reflecting better data utilization. Notably, CRFed shows the most significant gains, with accuracy increasing from 0.389 to 0.425 on CIFAR-100 and mAP from 0.882 to 0.920 on NIPD. This demonstrates CRFed's superior scalability and effectiveness in handling more edge nodes, making it robust in federated learning environments with increasing data sources.

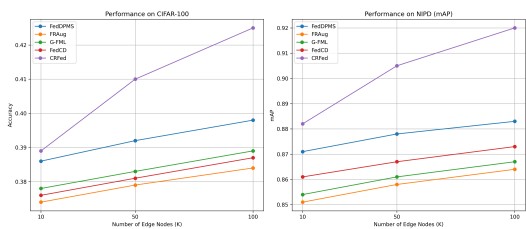

Figure 4: Effect of Increasing Edge Nodes

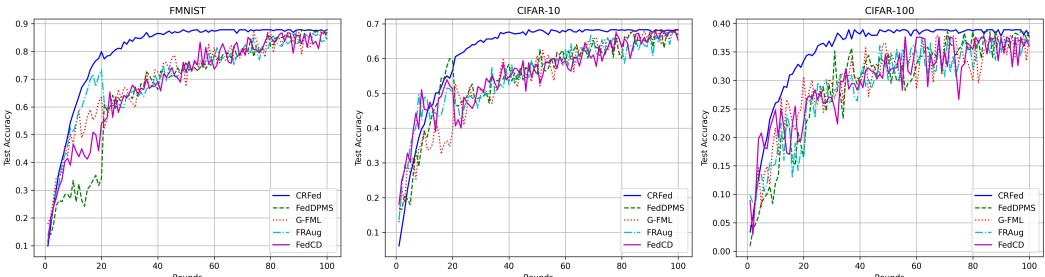

Figure 5: Test accuracy across federated training rounds for top 5 models on FMNIST, CIFAR-10, and CIFAR-100 datasets.

**Convergence Rate** The convergence performance of the top five models on FMNIST, CIFAR-10, and CIFAR-100 datasets is depicted in Figure 5. As observed, CRFed demonstrates significantly faster and more stable convergence compared to the competing methods across all three datasets. This superior performance is attributed to the diffusion-based data harmonization mechanism, which effectively aligns data distributions, and the confusion-resistant strategy that selects reliable clients and adaptively adjusts learning rates, ensuring efficient and robust training even in highly heterogeneous environments.

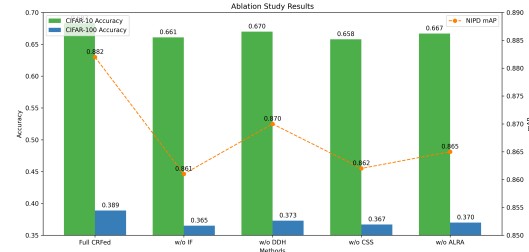

Figure 6: Ablation study results on CIFAR-10, CIFAR-100, and NIPD datasets. The bar charts show the accuracy on CIFAR-10 and CIFAR-100 datasets, while the line plot represents the mAP on the NIPD dataset.

## 4.3 Ablation Study

To evaluate the contribution of each component in the CRFed framework, we conduct an ablation study by removing or altering specific components and observing the impact on model performance. Removing the Indicator Function and using uniform sampling led to significant performance drops, with CIFAR-10 accuracy falling from 0.683 to 0.661, CIFAR-100 from 0.389 to 0.365, and NIPD mAP from 0.882 to 0.861. Excluding the Diffusion-based Data Harmonization (DDH) mechanism resulted in reduced accuracy on CIFAR-10 (0.670), CIFAR-100 (0.373), and NIPD mAP (0.870), highlighting its role in aligning data distributions. Replacing strategic client selection with random selection markedly decreased performance, emphasizing the importance of reliable client selection. Fixing the learning rate instead of adapting it slowed convergence and destabilized training. These findings validate the theoretical and practical significance of our proposed components in improving federated learning performance.

## 4.4 Comparison with Importance Sampling Methods

Previous importance sampling methods typically require prior analysis of the data relevance at each client-side [Hsu et al., 2020, Tian et al., 2022] or necessitate deriving optimal sampling weights based on assumptions such as the convexity of the loss function [Rizk et al., 2022, Zhu et al., 2024]. While these methods offer strong theoretical guarantees, they are somewhat limited in their adaptability to real-world federated learning (FL) scenarios. For instance, both FedIR [Hsu et al., 2020] and Harmony [Tian et al., 2022] assume that the server has knowledge of the local distributions of all clients. Although this assumption does not violate the privacy-preserving principles of FL, it can be challenging to obtain in real-world applications.

In contrast, our CRFed does not depend on these assumptions. Instead, it iteratively adjusts the data distributions during the FL process itself, enabling the model to dynamically harmonize the diverse, non-IID data across clients without requiring explicit distributional assumptions or centralized access to all client data distributions. Guided by the indicator function, our CRFed can derive the optimal sampling strategy for each local node.

Moreover, as shown in Table 3, empirical experiments demonstrate that the diffusion model achieves superior performance, outperforming other benchmark methods.

It is worth noting that this comparison is not entirely fair, as each importance sampling method operates under different assumptions. For example, ISFL requires a validation set to update the empirical gradient Lipschitz constants for each local model, while FedIR requires all clients to upload the conditional distribution of images given class labels to match the target distribution. Nevertheless, our CRFed outperforms the others even under less restrictive conditionsunlike ISFedAvg and ISFL, it does not require assumptions about the loss function or gradient variance, and unlike FedIR and Harmony, it does not require centralized access to all client data distributions before calculating the importance sampling weights.

Table 3: The performance of different importance sampling methods on CIFAR-100 under various $\beta$ values.

| | CIFAR-100 | | |
|---|---|---|---|
| **Method** | $\beta = 0.1$ | $\beta = 0.3$ | $\beta = 0.5$ |
| ISFedAvg | 0.232 | 0.285 | 0.305 |
| ISFL | 0.237 | 0.296 | 0.314 |
| FedIR | 0.258 | 0.311 | 0.352 |
| Harmony | 0.246 | 0.313 | 0.354 |
| CRFed | **0.280** | **0.345** | **0.389** |

## 5   Conclusion

In conclusion, this study tackles the pressing challenge of handling non-i.i.d. data in federated learning environments. We propose the Confusion-Resistant Federated Learning via Consistent Diffusion (CRFed) framework. This framework introduces a novel Indicator Function that dynamically adjusts sample weighting, facilitating a self-paced learning paradigm that prioritizes more difficult samples over time. Additionally, our diffusion-based data harmonization mechanism ensures consistent and aligned data distributions through iterative noise injection and denoising processes, mitigating the adverse effects of data heterogeneity. Our strategic client selection method, guided by the Indicator Function, ensures that the most reliable clients are chosen for training, thus improving the robustness and consistency of global model updates.

Despite the promising results, our approach has certain limitations. The reliance on complex diffusion mechanisms and adaptive strategies may introduce computational overhead, which could be a concern for resource-constrained environments. Future work should focus on optimizing the computational efficiency of the CRFed framework and exploring its applicability to a broader range of real-world federated learning scenarios [Zhang et al., 2023].

## 6   Acknowledgement

This research was funded by the Basic Science Center Project for National Natural Science Foundation of China (Grant No: 72088101), the Xiangjiang Laboratory Major Project(Grant No: 23XJ01007), and the Fundamental Research Funds for the Central Universities of Central South University(Grant No: 1053320214050).

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

# A  Appendix

## A.1  Supplementary Explanation of the Diffusion-based Data Harmonization Mechanism

**Forward Diffusion Process**  In the forward diffusion process, Gaussian noise with variance $\beta_t \in (0, 1)$ is added gradually to the sample $\mathbf{x}_0$ for $T$ steps. The process is defined as:

$$q(\mathbf{z}_{1:T}|z_i) = \prod_{t=1}^{T} q(\mathbf{z}_t|\mathbf{z}_{t-1}), \tag{16}$$

$$q(\mathbf{z}_t|\mathbf{z}_{t-1}) = \mathcal{N}(\mathbf{z}_t; \sqrt{1-\beta_t}\mathbf{z}_{t-1}, \beta_t\mathbf{I}). \tag{17}$$

Using notations $\alpha_t = 1 - \beta_t$ and $\bar{\alpha}_t = \prod_{s=1}^{t} \alpha_s$, the sample $\mathbf{z}_t$ can be defined directly as:

$$\mathbf{z}_t = \sqrt{\bar{\alpha}_t}z_i + \sqrt{1-\bar{\alpha}_t}\boldsymbol{\epsilon}, \qquad \boldsymbol{\epsilon} \sim \mathcal{N}(0, \mathbf{I}). \tag{18}$$

**Reverse Denoising Process**  The reverse denoising process aims to sample reversely from $\mathbf{z}_T$ through transition probabilities $q(\mathbf{z}_{t-1}|\mathbf{z}_t)$ for timesteps $T-1$ through 1 to obtain a sample drawn from $q(z_i)$. The transition $q(\mathbf{z}_{t-1}|\mathbf{z}_t)$ is a Gaussian distribution, tractable when conditioned on $z_i$:

$$q(\mathbf{z}_{t-1}|\mathbf{z}_t, z_i) = \mathcal{N}(\mathbf{z}_{t-1}; \tilde{\boldsymbol{\mu}}_t(\mathbf{z}_t, z_i), \tilde{\beta}_t\mathbf{I}), \tag{19}$$

where the mean $\tilde{\boldsymbol{\mu}}_t$ and variance $\tilde{\beta}_t$ are calculated as:

$$\tilde{\boldsymbol{\mu}}_t(\mathbf{z}_t, z_i) = \frac{1}{\sqrt{\alpha_t}}\left(\mathbf{z}_t - \frac{1-\alpha_t}{\sqrt{1-\bar{\alpha}_t}}\boldsymbol{\epsilon}_t\right), \tag{20}$$

$$\tilde{\beta}_t = \frac{1-\bar{\alpha}_{t-1}}{1-\bar{\alpha}_t}\beta_t. \tag{21}$$

The reverse transition probability $p_\theta(\mathbf{z}_{t-1}|\mathbf{z}_t)$ relies on the entire data distribution and is approximated through a neural network:

$$p_\theta(\mathbf{z}_{t-1}|\mathbf{z}_t) = N(\mathbf{z}_{t-1}; \boldsymbol{\mu}_\theta(\mathbf{z}_t, t), \boldsymbol{\Sigma}_\theta(\mathbf{z}_t, t)), \tag{22}$$

where $\boldsymbol{\Sigma}_\theta(\mathbf{z}_t, t) = \tilde{\beta}_t\mathbf{I}$ and the mean $\boldsymbol{\mu}_\theta(\mathbf{z}_t, t)$ depends on a noise sample $\boldsymbol{\epsilon}_\theta(\mathbf{z}_t, t)$ learned by a neural network. The learning process is guided by the objective function:

$$L = \mathbb{E}_{t, z_i, \epsilon}\left[||\boldsymbol{\epsilon} - \boldsymbol{\epsilon}_\theta(\mathbf{z}_t, t)||^2\right], \tag{23}$$

while the output sample is obtained as:

$$\mathbf{z}_{t-1} = \frac{1}{\sqrt{\alpha_t}}\left(\mathbf{z}_t - \frac{1-\alpha_t}{\sqrt{1-\bar{\alpha}_t}}\boldsymbol{\epsilon}_\theta(\mathbf{z}_t, t)\right) + \sigma_t\mathbf{z}, \tag{24}$$

where $\mathbf{z} \sim \mathcal{N}(0, \mathbf{I})$ if $t > 1$ and $\mathbf{z} = 0$ otherwise.

## A.2  Proof of Theorem 3.1

*Proof.*  For convenience, define $\omega_i = \omega_i(\theta_t) = \sigma_i^* + (l_i - \tau)\,k + 2\,\lambda\,\frac{\log\sigma_i^*}{\sigma_i^*}\,k$, ,which can be regarded as the effective weight term induced by the indicator function $I_\lambda(l_i, \sigma_i)$. It appears in the local update of client $i$. Hence, the update rule can be written as:

$$\theta_{t+1} = \theta_t - \eta\,\omega_i\,\nabla_\theta l_i. \tag{25}$$

Meanwhile, we assume the local loss function $l_i(\theta)$ satisfies two common assumptions:

1. **$L$-smoothness (Lipschitz continuity of gradients):** The gradient of $l_i(\theta)$ is $L$-Lipschitz, i.e.,

$$\|\nabla l_i(\theta_1) - \nabla l_i(\theta_2)\| \; \leq \; L \, \|\theta_1 - \theta_2\| \quad \text{for all } \theta_1, \theta_2. \tag{26}$$

2. **Bounded below:** There exists a constant $l_i^*$ such that

$$l_i(\theta) \; \geq \; l_i^* \tag{27}$$

for all $\theta$.

From $L$-smoothness, we have for the update from iteration $t$ to $t+1$:

$$l_i(\theta_{t+1}) \; \leq \; l_i(\theta_t) \; + \; \langle \nabla l_i(\theta_t), \, \theta_{t+1} - \theta_t \rangle \; + \; \frac{L}{2} \|\theta_{t+1} - \theta_t\|^2. \tag{28}$$

Using (25), we get

$$\theta_{t+1} - \theta_t \; = \; -\eta \, \omega_i \, \nabla_\theta l_i(\theta_t). \tag{29}$$

Substituting back yields

$$l_i(\theta_{t+1}) \; \leq \; l_i(\theta_t) \; - \; \eta \, \omega_i \, \|\nabla l_i(\theta_t)\|^2 \; + \; \frac{L}{2} \, \eta^2 \, \omega_i^2 \, \|\nabla l_i(\theta_t)\|^2$$

$$= \; l_i(\theta_t) \; - \; \left( \eta \, \omega_i \; - \; \frac{L}{2} \, \eta^2 \, \omega_i^2 \right) \|\nabla l_i(\theta_t)\|^2. \tag{30}$$

If we ensure $\eta \, \omega_i \; \leq \; \frac{1}{L}$, or more conservatively $\eta \, \omega_i \; \leq \; \frac{1}{2L}$, then

$$\eta \, \omega_i \; - \; \frac{L}{2} \, \eta^2 \, \omega_i^2 \; \geq \; \frac{1}{2} \, \eta \, \omega_i. \tag{31}$$

Hence, (30) implies

$$l_i(\theta_{t+1}) \; \leq \; l_i(\theta_t) \; - \; \frac{1}{2} \, \eta \, \omega_i \, \|\nabla l_i(\theta_t)\|^2. \tag{32}$$

Therefore, if $\eta$ is chosen such that $\eta \, \omega_i \leq \frac{1}{2L}$, then $l_i(\theta)$ is guaranteed to decrease at every step, forming a monotonically decreasing sequence.

By the bounded-below assumption $l_i(\theta) \geq l_i^*$, the loss cannot decrease indefinitely. Consequently, $\{ l_i(\theta_t) \}$ converges. From a standard telescoping sum argument, at iteration $T$, we estimate the sum of squared gradient norms:

$$\sum_{t=0}^{T-1} \|\nabla l_i(\theta_t)\|^2 \; \leq \; \frac{2}{\eta} \sum_{t=0}^{T-1} \frac{l_i(\theta_t) - l_i(\theta_{t+1})}{\omega_i} \; \leq \; \frac{2 \left[ l_i(\theta_0) - l_i^* \right]}{\eta \, \min_t \{ \omega_i(\theta_t) \}}. \tag{33}$$

If $\min_t \{ \omega_i(\theta_t) \} > 0$ and $\eta$ is properly chosen, then the average gradient norm goes to 0 as $T \to \infty$, ensuring global convergence.

In FedAvg, the single-step update typically has the form $\theta_{t+1} = \theta_t - \eta_{\text{FedAvg}} \nabla_\theta l_i(\theta_t)$. In CRFed, we have an additional factor $\omega_i(\theta_t)$. Thus the effective learning rate becomes $\eta_{\text{eff}} = \eta \, \omega_i(\theta_t)$. If $\omega_i(\theta_t) \leq 1$ (which can be ensured by proper design of the indicator function in many scenarios), then $\eta_{\text{eff}} \leq \eta$, i.e., CRFed is more conservative (or adaptive) in its step size. This helps avoid large updates caused by data heterogeneity and promotes more stable convergence. Consequently, CRFed obtains a tighter convergence bound because it mitigates the gradient-directional deviation brought on by heterogeneous data.

$\square$

## A.3 Proof of Theorem 3.2

*Proof.* Recall that the Indicator Function is defined as $I_\lambda(l_i, \sigma_i) = (l_i - \tau)\,\sigma_i + \lambda\,(\log \sigma_i)^2.$, we wish to find the optimal $\sigma_i^*$ that minimizes this function for a given $l_i$.

To simplify the differentiation, we introduce two transformations:

$$c_i = \frac{l_i - \tau}{\lambda} \quad \text{and} \quad x_i = \log \sigma_i. \tag{34}$$

Hence, we have

$$\sigma_i = e^{x_i}. \tag{35}$$

Under these new variables, the Indicator Function becomes

$$I_\lambda(l_i, \sigma_i) = (l_i - \tau)\,e^{x_i} + \lambda\,(x_i)^2 = \lambda\Big(c_i\,e^{x_i} + x_i^2\Big). \tag{36}$$

Since $\lambda > 0$ is just a constant multiplier, minimizing $I_\lambda$ is equivalent to minimizing

$$f(x_i) = c_i\,e^{x_i} + x_i^2. \tag{37}$$

We now compute the derivative of $f(x_i)$ with respect to $x_i$ and set it to zero to find the critical points:

$$\frac{\mathrm{d}}{\mathrm{d}x_i}\big[\,c_i\,e^{x_i} + x_i^2\,\big] = c_i\,e^{x_i} + 2\,x_i = 0. \tag{38}$$

Rearrange this to isolate exponential terms:

$$c_i\,e^{x_i} = -2\,x_i. \tag{39}$$

Note that this step implicitly assumes $x_i < 0$ if the right-hand side is negative, depending on the sign of $c_i$. We proceed to manipulate this into a standard Lambert W form. Multiply both sides by $-\frac{1}{2}\,e^{x_i}$:

$$-\tfrac{1}{2}\,c_i\,e^{2\,x_i} = x_i\,e^{x_i}. \tag{40}$$

Let

$$y = x_i\,e^{x_i}. \tag{41}$$

Then,

$$y = -\tfrac{1}{2}\,c_i\,e^{2\,x_i}. \tag{42}$$

Meanwhile, by definition of $y$,

$$y = x_i\,e^{x_i}. \tag{43}$$

Hence, we arrive at

$$x_i = -W\Big(\tfrac{c_i}{2}\Big) \quad \Longrightarrow \quad x_i = -W\Big(\tfrac{l_i - \tau}{2\,\lambda}\Big), \tag{44}$$

where $W(\cdot)$ is the Lambert W function, the inverse function of $z \mapsto z\,e^z$.

In practice, the argument of the Lambert W function must lie in a domain where the function is real-valued. By restricting

$$l_i - \tau \geq -\tfrac{2}{e}, \tag{45}$$

we ensure that

$$\frac{l_i - \tau}{2\,\lambda} \geq -\tfrac{1}{e}. \tag{46}$$

Therefore, to handle the case when $l_i - \tau < -\tfrac{2}{e}$, we take

$$\max\Big(-\tfrac{2}{e},\, l_i - \tau\Big), \tag{47}$$

which ensures the Lambert W functions argument stays within the valid real domain. Consequently, the solution for $x_i$ becomes

$$x_i = -W\Big(\tfrac{1}{2\lambda}\,\max\big(-\tfrac{2}{e},\, l_i - \tau\big)\Big). \tag{48}$$

Recalling $\sigma_i = e^{x_i}$, we conclude:

$$\sigma_i^*(l_i) = \exp\Big(-W\big(\tfrac{1}{2\lambda}\,\max\big(-\tfrac{2}{e},\, l_i - \tau\big)\big)\Big). \tag{49}$$

$\square$

## A.4 Design and Training Details of the Model Encoder and Distribution Decoder

In the CRFed framework, both the model encoder and distribution decoder play crucial roles in ensuring effective data harmonization and robust model updates. These components are implemented using autoencoder architectures, designed to compress and reconstruct data representations efficiently.

### A.4.1 Model Encoder

The model encoder $E$ is responsible for compressing the global model parameters $\theta_t$ into a lower-dimensional meta-model representation $\phi_t$. The encoder architecture comprises several fully connected layers activated by ReLU functions, followed by a linear transformation layer to produce the final compressed representation.

Mathematically, given the input global model parameters $\theta_t \in \mathbb{R}^d$, the encoder outputs a compressed representation $\phi_t = E(\theta_t) \in \mathbb{R}^{d'}$, where $d' < d$. The transformation is defined as follows:

$$h_1 = \text{ReLU}(W_1\theta_t + b_1) h_2 = \text{ReLU}(W_2 h_1 + b_2) \vdots h_k = \text{ReLU}(W_k h_{k-1} + b_k) \phi_t = W_{out} h_k + b_{out} \tag{50}$$

The training of the model encoder involves minimizing the mean squared error (MSE) between the original model parameters and their reconstructions. The loss function is given by:

$$\mathcal{L}_E = \frac{1}{N} \sum_{i=1}^{N} \|\theta_{t_i} - \hat{\theta}_{t_i}\|^2 \tag{51}$$

where $\hat{\theta}_{t_i} = E^{-1}(E(\theta_{t_i}))$ and $N$ is the number of samples. The optimization is performed using the Adam optimizer with a learning rate $\eta$. The detailed architecture includes an input layer of size $d$, hidden layers of sizes $[128, 64, 32]$, and an output layer of size $d' = 16$.

### A.4.2 Distribution Decoder

The distribution decoder $D$ aims to transform the denoised latent representations $\mathbf{z}_t$ back into the desired data distribution. Like the encoder, the decoder uses a series of fully connected layers with ReLU activations, culminating in a linear layer to reconstruct the data.

Given the input latent representation $\mathbf{z}_t \in \mathbb{R}^{d'}$, the decoder outputs the reconstructed data $\hat{\mathbf{x}}_t = D(\mathbf{z}_t) \in \mathbb{R}^d$. The transformations are defined as:

$$h_1 = \text{ReLU}(W_1'\mathbf{z}_t + b_1') h_2 = \text{ReLU}(W_2' h_1 + b_2') \vdots h_k = \text{ReLU}(W_k' h_{k-1} + b_k') \hat{\mathbf{x}}_t = W_{out}' h_k + b_{out}' \tag{52}$$

The training process for the distribution decoder also minimizes the MSE, defined as:

$$\mathcal{L}_D = \frac{1}{N} \sum_{i=1}^{N} \|\mathbf{x}_{t_i} - \hat{\mathbf{x}}_{t_i}\|^2 \tag{53}$$

where $\hat{\mathbf{x}}_{t_i} = D(\mathbf{z}_{t_i})$ and $N$ is the number of samples. The optimization employs the Adam optimizer with a learning rate $\eta$. The architecture details include an input layer of size $d' = 16$, hidden layers of sizes $[32, 64, 128]$, and an output layer of size $d$.

### A.4.3 Architectural Details

The model encoder and distribution decoder share a similar architectural approach, emphasizing efficient compression and reconstruction through deep learning techniques. The key parameters for both autoencoders are summarized as follows:

- **Model Encoder:**
    - Input layer size: $d$
    - Hidden layer sizes: $[128, 64, 32]$
    - Output layer size: $d' = 16$

- – Activation function: ReLU
- – Learning rate: $\eta = 0.001$
- – Batch size: 32
- **Distribution Decoder:**
  - – Input layer size: $d' = 16$
  - – Hidden layer sizes: $[32, 64, 128]$
  - – Output layer size: $d$
  - – Activation function: ReLU
  - – Learning rate: $\eta = 0.001$
  - – Batch size: 32

These architectural and training details ensure that the CRFed framework can effectively handle non-i.i.d. data distributions, facilitating robust and consistent model updates across federated learning environments.

## A.5 Pseudocode for CRFed

The complete computational process of CRFed is illustrated in Algorithm 1.

---

**Algorithm 1** Confusion-Resistant Federated Learning via Consistent Diffusion (CRFed)

---

**Require:** Maximum global rounds $T_G$, local training cycles $E_l$, client weights $\{\pi_k\}$, local datasets $\{\mathcal{D}_k\}$, learning rate $\eta_0$, indicator function threshold $\gamma$
1: Initialize global model parameters $\theta_0$
2: Initialize local model parameters $\{\theta_0^k\}$ and importance sampling weights $\{w_i^k \leftarrow 1\}$
3: Set $t \leftarrow 1$
4: **while** $t \leq T_G \times E_l$ **do**
5:     **for** each client $k$ **do**
6:         Sample local data $\mathcal{D}_i^{\text{sampled}}$ based on importance weights $w_i^k$
7:         Train local model $\theta_t^k$ on $\mathcal{D}_i^{\text{sampled}}$
8:     **end for**
9:     **if** $t \mod E_l == 0$ **then**
10:         Each client uploads local model $\{\theta_t^k\}$ to the server
11:         Server aggregates the global model: $\bar{\theta}_t \leftarrow \sum_{k=1}^{K} \pi_k \theta_t^k$
12:         **for** each client $k$ **do**
13:             Compute optimal indicator function $I^*$
14:             Calculate optimal sampling probability $P_i^* = \text{ReverseDenoise}(I^*)$
15:             Calculate importance sampling weights $w_i = \frac{P_i^*}{P_0}$
16:             Sample new local data $\mathcal{D}_i^{\text{sampled}}$ based on updated importance weights $w_i$
17:             Train local model $\theta_t^k$ on $\mathcal{D}_i^{\text{sampled}}$
18:             Update local model $\theta_t^k \leftarrow \bar{\theta}_t$
19:             Adjust learning rate: $\eta_i = \eta_0 \cdot \frac{I_\lambda(l_i, \sigma_i)}{\max_j I_\lambda(l_j, \sigma_j)}$
20:         **end for**
21:     **end if**
22:     $t \leftarrow t + 1$
23: **end while**
24: **Output:** Global model $\bar{\theta}_t$, local models $\{\theta_t^k\}$

---

## A.6 Detailed Model Structure

The detailed configuration of the models used in our experiments is provided below. Each table outlines the layers and parameters for the respective datasets.

All weights are initialized with a normal distribution (mean 0, standard deviation 0.1) and biases with a constant value of 0.1. These settings ensure that the models are well-prepared for training and capable of achieving high performance on the respective datasets.

Table 4: Model structure for MNIST and FashionMNIST datasets

| Layer | Type | Output Channels/Units | Additional Information |
|---|---|---|---|
| Input | - | - | 28x28 grayscale images |
| 1 | Convolutional | 16 | 5x5 filters, stride 1, padding 'SAME' |
| - | Activation | - | ReLU |
| - | Max Pooling | - | 2x2 window, stride 2 |
| 2 | Convolutional | 32 | 5x5 filters, stride 1, padding 'SAME' |
| - | Activation | - | ReLU |
| - | Max Pooling | - | 2x2 window, stride 2 |
| 3 | Fully Connected | 512 | - |
| - | Activation | - | ReLU |
| 4 | Fully Connected | 10 | Softmax |

Table 5: Model structure for CIFAR-10 and CIFAR-100 datasets

| Layer | Type | Output Channels/Units | Additional Information |
|---|---|---|---|
| Input | - | - | 32x32 RGB images |
| 1 | Convolutional | 64 | 5x5 filters, stride 1, padding 'SAME' |
| - | Activation | - | ReLU |
| - | Max Pooling | - | 2x2 window, stride 2 |
| 2 | Convolutional | 64 | 5x5 filters, stride 1, padding 'SAME' |
| - | Activation | - | ReLU |
| - | Max Pooling | - | 2x2 window, stride 2 |
| 3 | Fully Connected | 1600 | - |
| - | Activation | - | ReLU |
| 4 | Fully Connected | 512 | - |
| - | Activation | - | ReLU |
| 5 | Fully Connected | 10 (CIFAR-10) / 100 (CIFAR-100) | Softmax |

In the NIPD dataset, we adopted the classic YOLOv3 [Redmon and Farhadi, 2018] model.

