# OpenReview forum: "Confusion-Resistant Federated Learning via Diffusion-Based Data Harmonization on Non-IID Data"
_NeurIPS.cc/2024/Conference — NeurIPS 2024 poster_

### Official Review · Reviewer_CpsC · 2024-07-09

**Soundness:** 2
**Presentation:** 1
**Contribution:** 2
**Rating:** 4
**Confidence:** 4

**Summary:**

This paper proposes an importance sampling method with a diffusion model to achieve data harmonization in federated learning with non-i.i.d. data. The proposed method utilizes the indicator function from self-paced learning to measure the reliability of loss on each client and calculates the optimal data distribution according to the indicator function. The proposed method is evaluated in five different datasets empirically.

**Strengths:**

This paper proposes a novel method to measure the importance and difficulty of each data sample, and samples the local data with importance sampling to achieve lower model update divergence. Experiments are conducted on five different datasets, with a comparison to ten other baselines, sufficiently showing the superior performance of the proposed method.

**Weaknesses:**

Despite the sufficient experimental results, the poor clarity of this paper weakens its soundness. There are several confusing points in the context of this paper:

1.	It is unclear what the indicator function means and how it is derived. As the core of this paper, the indicator function is only provided with a formulation and never explained how it is derived. Additionally, it is not straightforward to understand the relationship between the indicator function and the optimal data distribution.

2.	How the diffusion model is trained and used is not discussed in the paper. What we know from the paper is that the diffusion model takes the model embedding and the indicator function together to generate the optimal data distribution. It is weird why the diffusion model is necessary here. It seems that what we need is only a method that can estimate the optimal data distribution according to the indicator function.

3.	The definitions of the indicator function in Eq. (1) and Eq. (16) seem to be inconsistent. The indicator function is calculated for each sample in Eq. (1), while it is calculated for each client in Eq. (16).

After all, the core idea of this paper is simple: re-assigning sampling probability to achieve overall balanced training data distribution such that the model update divergence can be mitigated. It is doubtful whether such a complicated framework is necessary to achieve this goal, while the improvement seems to be limited in the final converged accuracy.

**Questions:**

1.	Why is the optimal uncertainty obtained by minimizing the indicator function?

2.	How is the distribution decoder trained? Since the decoder should output an unknown distribution from the denoised latent representation, there should not be a ground truth $x_{t_i}$ when training the decoder.

3.	Are the diffusion model, the model encoder, and the distribution decoder trained locally or globally? What is the training overhead of these components?

**Limitations:**

See the weaknesses and questions above.

---

> ### Author Rebuttal · Authors · 2024-08-07
>
> # W1. Discussion on Indicator Function.
>
>  **Response:** We appreciate the reviewer's insightful comments. Below is a comprehensive explanation:
>
> (1) **Motivation of the Indicator Function**.   The Indicator Function $I_{\lambda}(l_i, \sigma_i)$ is designed to dynamically adjust sample weighting based on loss values and uncertainties, **inspired by self-paced learning** [1]. It follows curriculum learning principles, prioritizing simpler tasks before more complex ones, to improve training convergence efficiency. **The proof of this assertion can be found in global rebuttal**.
>
> (2) **Derivation of the Indicator Function**. Our design is heuristic, similar to confidence-aware cross-entropy [2]. It consists of a loss-amplifying term and a regularization term, enhancing high-loss sample contributions while regularizing uncertainty estimation. To further discussion, see the convergence speed discussion of CRFed with FedAvg in the global rebuttal.
>
> (3)  **The Relationship Between the Indicator Function and the Optimal Data Distribution.** Please refer to the **response to Q1** for detailed explanation.
>
> # W2. Necessity of diffusion model
>
>  **Response:** Thanks for your comment. In CRFed, the diffusion-based data harmonization process is integrated into each global aggregation round. This means the harmonization update occurs every time the global model is aggregated and broadcasted to the clients, specifically set to once per global round. The model is trained online during the federated learning process.
>
> To address the reviewer's concern, we conducted comparative experiments with alternative methods to estimate the optimal data distribution based on the indicator function:
>
> - **Simple Weighted Sampling (SWS)**: Directly adjusts sampling probabilities based on indicator function values without additional transformation.
> - **Kernel Density Estimation (KDE)**: Uses KDE to smooth and adjust data distribution based on the indicator function.
>
> The results, shown in **Table 8 on the one-page pdf**, indicate the diffusion model achieves the best performance. **The diffusion model's iterative denoising process effectively harmonizes data distributions**, ensuring robust and consistent alignment, crucial for improving convergence and performance.
>
> From a practical perspective, CRFed harmonizes these data distributions, ensuring consistent updates and improved convergence. CRFed outperforms state-of-the-art methods and remains effective, as its training can be performed during the local phase(although running on server), ensuring real-time operation of the FL system.
>
> # W3. Issue about Eq. (1) and Eq. (16)
>
>  **Response:** : We appreciate the reviewer's observation. The indicator function is defined at the client level but computed from the sample level. Specifically, Eq. (1) calculates the indicator function for each sample, while Eq. (16) aggregates it at the client level. There is no inconsistency between Eq. (1) and Eq. (16). We will improve clarity in future version.
>
> # Q1. Why is the optimal uncertainty obtained by minimizing the indicator function?
>
> **Response:** Thank you for your comment. Minimizing the indicator function is crucial for:
>
> - **Balancing Loss and Uncertainty:** Minimizing $I_{\lambda}(l_i, \sigma_i)$ ensures higher importance for samples with higher loss values while controlling uncertainty.
> - **Preventing Overfitting:** The regularization term $\lambda (\log \sigma_i)^2$ discourages high uncertainties, preventing overfitting to unreliable samples.
>
> Additionally, minimizing the indicator function helps control the parameter update magnitude, as shown by:
>
> $$
> \left| \theta_{t+1} - \theta_t \right| = \eta \left| \left( \sigma_i^* + (l_i - \tau) k + 2\lambda \frac{\log \sigma_i^*}{\sigma_i^*} k \right) \nabla_{\theta} l_i \right| \leq \eta C \left| \nabla_{\theta} l_i \right|,
> $$
>
> where $\eta$ is the learning rate, and $C$ is a constant that bounds the term $\left( \sigma_i^* + (l_i - \tau) k + 2\lambda \frac{\log \sigma_i^*}{\sigma_i^*} k \right)$. This inequality indicates that by choosing $\sigma_i^*$ to minimize the indicator function, we can control the magnitude of the parameter updates.
>
> # Q2. Discussion on the distribution decoder
>
> **Response:** Thank you for your comment. The role of the Distribution Decoder is essentially to map the latent representation $z_t$ to the distribution space, so the training is conducted offline on a known ground truth distribution. For a specific task, using the task's test set (or validation set) is a good choice. Other details can be found in section A.2.2 of the original text. We hope these explanations can address your concern.
>
> #  Q3. Details of model training
>
> **Response:** Thank you for your comment. The diffusion model, the model encoder, and the distribution decoder are all trained globally. Specifically, the model encoder and the distribution decoder are trained offline.
>
> To quantify the training overhead introduced by our approach, we have measured the wall-clock time per round and the total convergence time. The results are shown in **Table 2 on the one-page pdf**. We observe that the wall-clock time per round for CRFed is slightly higher compared to other methods due to the additional noise/denoise operations, but the total convergence time is comparable to other methods. The slight increase in computational cost is balanced by the improved model performance, making it a worthwhile trade-off.
>
> ## Reference
>
> [1]  Self-paced learning: An implicit regularization perspective. AAAI, 2017.
>
> [2] Data parameters: A new family of parameters for learning a differentiable curriculum. Neurips, 2019.

---

> > ### Comment · Reviewer_CpsC · 2024-08-07
> >
> > I appreciate the response from the author, which helps me better understand the methodology in the paper. However, I am still concerned about the motivation for using the diffusion model for data importance sampling. The additional computation and data required for training the diffusion model offline are still unclear. And the comparison with other importance sampling methods for FL, e.g., FedIR (Federated Visual Classification with Real-World Data Distribution) and Harmony (HARMONY: Heterogeneity-Aware Hierarchical Management for Federated Learning System), is absent. Accordingly, I would increase my score to 4.

---

> ### Author Response · Authors · 2024-08-12
> **Further discussions on motivation and computational costs**
>
> Thank you for your timely feedback and for raising the score. We’ve provided the following response to address the remaining questions you raised:
>
> ### 1. Motivation for Diffusion Model
> Previous importance sampling methods typically require prior analysis of the data relevance at each client-side [3,4] or necessitate deriving optimal sampling weights based on assumptions such as the convexity of the loss function [1,2]. While these methods offer strong theoretical guarantees, they are somewhat limited in their adaptability to real-world FL scenarios. For instance, both FedIR[3] and Harmony[4] assume that the server has knowledge of the local distributions of all clients. Although this assumption does not violate the privacy-preserving principles of FL, it can be challenging to obtain in real-world applications.
>
> In contrast, the diffusion model based method we proposed does not depend on these assumptions. Instead, it iteratively adjusts the data distributions during the FL process itself. This enables the model to dynamically harmonize the diverse, non-IID data across clients **without requiring explicit distributional assumptions or centralized access to all client data distributions**. Guided by the indicator function, our CRFed can derive the optimal sampling strategy for each local node.
>
> Moreover, as shown in the table below, empirical experiments demonstrate that **the diffusion model achieves superior performance, outperforming other benchmark methods**.
>
> #### The performance of different importance sampling methods on CIFAR-100 under various β values.
>
> | Method   | β=0.1 | β=0.3 | β=0.5 |
> |----------|-------|-------|-------|
> | ISFedAvg | 0.232 | 0.285 | 0.305 |
> | ISFL     | 0.237 | 0.296 | 0.314 |
> | FedIR    | 0.258 | 0.311 | 0.352 |
> | Harmony  | 0.246 | 0.313 | 0.354 |
> | CRFed    | **0.280** | **0.345** | **0.389** |
>
> It’s worth noting that this comparison isn’t entirely fair, as each importance sampling method operates under different assumptions. For example, ISFL requires a validation set to update the empirical gradient Lipschitz constants for each local model, while FedIR requires all clients to upload the conditional distribution of images given class labels that matches the target distribution. Nevertheless, our **CRFed outperforms the others even under less restrictive conditions**—unlike ISFedAvg and ISFL, it doesn’t require assumptions about the loss function or gradient variance, and unlike FedIR and Harmony, it doesn’t require centralized access to all client data distributions before calculating the importance sampling weights.
>
> ### 2. Additional Computation
> We acknowledge that CRFed requires additional computational resources, primarily on the server side. However, in FL, the main computational burden typically lies in the local model training and communication. In practical FL systems, **increasing server-side computation to achieve performance gains is often desirable**, as it does not compromise the real-time operation of the FL system and tends to offer stronger economic benefits. For example, in CLIP2FL[5], to mitigate data heterogeneity and class imbalance, the server generates federated features based on client-uploaded gradients and uses CLIP's text encoder for prototype contrastive learning. Similarly, in FedMRUR[6], the server must compute compressed data received from clients, calculate the corresponding logits, and perform global knowledge matching, which involves substantial computational intensity.
>
> Lastly, we would like to express our sincere gratitude to Reviewer CpsC. Your feedback has inspired us to delve deeper into discussions regarding CRFed, and the additional experiments have further strengthened its contributions. We plan to include these results and discussions in future versions. We hope this response addresses some of your concerns.^_^
>
> ### References
> [1] **ISFedAvg**. Federated learning under importance sampling. IEEE Transactions on Signal Processing. 2022.
>
> [2] **ISFL**. Federated Learning for Non-iid Data with Local Importance Sampling. IEEE Internet of Things Journal. 2024
>
> [3] **FedIR**. Federated Visual Classification with Real-World Data Distribution. ECCV. 2020.
>
> [4] **Harmony**. Heterogeneity-aware hierarchical management for federated learning system. MICRO. 2022.
>
> [5] **CLIP2FL**. CLIP-Guided Federated Learning on Heterogeneity and Long-Tailed Data. AAAI. 2024.
>
> [6] **FedMRUR**. Federated learning with manifold regularization and normalized update reaggregation. Neurips. 2023.

---

### Official Review · Reviewer_FLG7 · 2024-07-10

**Soundness:** 3
**Presentation:** 3
**Contribution:** 3
**Rating:** 7
**Confidence:** 4

**Summary:**

This paper presents a framework called CRFed to address the significant challenges posed by non-i.i.d. data in federated learning environments. This work introduces a diffusion-based data harmonization mechanism that effectively reduces disparities in data distributions across different nodes. Additionally, the paper proposes a confusion-resistant strategy that leverages an adaptive indicator function based on importance sampling.
Overall, this paper's writing is clear and easy to follow. The figures are well-drawn, allowing for a quick understanding of the research motivation and methodological design. The core contribution of this paper is the introduction of a diffusion-based data harmonization method to obtain valuable data distribution for global model aggregation, which is a very brave and innovative idea. Additionally, how to use synthetic data to enhance training effectiveness has long been an open question in the field of FL, and this work clearly provides a very promising approach. Therefore, I recommend that this paper be accepted.

**Strengths:**

1. The method described in the paper is presented very clearly, the formulas are well-expressed, and the charts are clear.
2. The diffusion-based data harmonization mechanism is especially creative. This method uses Gaussian noise injection and iterative denoising to gradually align local data distributions with a desired global distribution. This process is clearly explained in Equations (7) to (12) and greatly reduces the impact of data differences. The detailed explanation of the forward and reverse processes, as shown in Figure 2, highlights a smart and practical way to handle non-i.i.d. data.
3.  The results in Tables 1 and 2 consistently show that CRFed outperforms other methods in both accuracy and convergence speed. The detailed comparison with various state-of-the-art methods under different $\beta$ values and edge node configurations highlights the practical applicability and scalability of the proposed framework.

**Weaknesses:**

1. Some notations could be more clearly defined. For instance, in the definition of the Indicator Function $ I_\lambda (l_i, \sigma_i) $, it's not immediately clear how $\tau$ (the confidence threshold) is dynamically adjusted or chosen. A more detailed explanation of how $\tau$ impacts the learning process and its optimal selection criteria would be beneficial.
2. The iterative denoising process might face scalability issues with very large datasets or a high number of iterations. The paper should evaluate the performance and efficiency of the denoising steps in such scenarios to understand the method’s scalability better.
3. It would be beneficial to include a brief discussion on future work in the paper.

**Questions:**

See above.

**Limitations:**

The authors have thoroughly outlined the limitations of their work as well as the potential negative societal impacts.

---

> ### Author Rebuttal · Authors · 2024-08-07
>
> # W1. Some notations could be more clearly defined.
>
> **Response:** Thanks for pointing this out. In our framework, $\tau$ represents a confidence threshold that determines the difficulty level of samples based on their loss values. The term $(l_i - \tau) \sigma_i$ in the Indicator Function adjusts the weight of each sample based on its loss value $l_i$ relative to the threshold $\tau$. This adjustment mechanism prioritizes samples with higher loss values (more difficult samples) when $l_i > \tau$, giving them higher weights, while samples with lower loss values (easier samples) are given lower weights. The impact of $\tau$ on the learning process is twofold:
>
> - By setting an appropriate $\tau$, the model can focus on learning from more difficult samples first, thereby implementing a form of curriculum learning. This helps the model to progressively handle more complex patterns in the data, leading to improved generalization and robustness.
> - $\tau$ can be dynamically adjusted during the training process to reflect the model's evolving understanding of the data. Initially, $\tau$ can be set to a lower value to focus on easier samples, and as training progresses, $\tau$ can be increased to prioritize more difficult samples. This dynamic adjustment ensures that the model continually challenges itself, preventing stagnation and promoting continuous improvement.
>
> # W2. Evaluation the performance and efficiency of the denoising steps.
>
> **Response:** We appreciate the reviewer's concern regarding the scalability of the iterative denoising process, especially when dealing with very large datasets or a high number of iterations. To address this, we have conducted additional experiments to evaluate the performance and efficiency of the denoising steps under such scenarios.
>
> We conducted experiments on the CIFAR-100 and NIPD datasets, varying the dataset size and the number of iterations in the denoising process. The key parameters evaluated include:
>
> - Dataset sizes: Full CIFAR-100 (50,000 samples) and NIPD (80,000 samples)
> - Number of denoising iterations: 10, 50, 100, 200
>
> The results  are shown in **Table 7 on the one-page pdf**. We can find that the performance improvements (in terms of accuracy for CIFAR-100 and mAP for NIPD) saturate as the number of iterations increases. These results demonstrate that while the iterative denoising process is effective, the benefits in performance diminish beyond a certain number of iterations. Therefore, for practical applications, we recommend limiting the number of denoising iterations to around 100, where a good balance between performance and efficiency is achieved. This approach ensures that the method remains scalable even for large datasets.
>
> # W3. A brief discussion on future work.
>
> **Response:**  Thanks for the suggestion. Future work could explore more sophisticated noise injection techniques in the diffusion-based data harmonization process. For instance, adaptive noise schemes that consider the specific characteristics of local data distributions could potentially improve the alignment of data across clients. We will include relevant discussions in future version to inspire further work.

---

### Official Review · Reviewer_cbew · 2024-07-11

**Soundness:** 4
**Presentation:** 3
**Contribution:** 3
**Rating:** 7
**Confidence:** 4

**Summary:**

This paper introduces CRFed, a framework designed to handle the challenges of non-i.i.d. data in federated learning. By using a diffusion-based data harmonization mechanism and a confusion-resistant strategy, CRFed aims to reduce data distribution differences among participating nodes and improve model consistency. Extensive experiments show that CRFed significantly enhances accuracy and convergence speed compared to existing methods.
In my opinion, the diffusion-based data harmonization mechanism is an innovative approach to dealing with data distribution differences. This paper not only introduces new theoretical concepts but also validates them through comprehensive experiments, making it a significant contribution to the field.

**Strengths:**

The paper exhibits several strengths that highlight its contributions and impact on the field of federated learning:
1. The introduction of a diffusion-based data harmonization mechanism is a fresh approach to tackling data distribution disparities. I think this idea has great potential to improve the stability of learning despite client heterogeneity. There's a mapping relationship between local and global data distributions, and using a diffusion model to capture this relationship is really innovative.
2.The paper conducts extensive experiments on various non-i.i.d. datasets, including MNIST, FashionMNIST, CIFAR-10, CIFAR-100, and NIPD. Especially with NIPD, which is a very challenging dataset, I congratulate the authors for achieving impressive performance on it.
3.The framework's ability to handle an increasing number of edge nodes and its adaptive learning rate adjustment contribute to improved scalability and training efficiency. This makes CRFed a practical solution for real-world federated learning scenarios.

**Weaknesses:**

There are some areas where the paper could be improved to enhance its clarity, robustness, and applicability:
1.The method involves several hyperparameters, like the variance of Gaussian noise, the regularization coefficient, and the dynamically adjusted confidence threshold. A more thorough sensitivity analysis of these parameters would make the paper more complete.
2.In cases where data distributions are extremely diverse or have intrinsic properties that are difficult to capture through these processes, the effectiveness of the harmonization might be limited.
3.While the method aims to enhance model consistency and reduce data disparities, the additional communication overhead introduced by the diffusion mechanism and importance sampling is not discussed.
4.Theorem 3.1 is valuable. However, during the derivation process, the purpose of equation (3) needs to be clearly explained.
5."I suggest highlighting the data showing performance advantages in Table 2 in bold.

**Questions:**

Please refer to the weaknesses.

**Limitations:**

The limitations are discussed in the paper by the authors. There is no potential negative societal impact.

---

> ### Author Rebuttal · Authors · 2024-08-07
>
> # W1. Sensitivity analysis.
>
> **Response:** We appreciate the reviewer's insightful comment regarding the sensitivity analysis of our hyperparameters. To address this, we conducted additional experiments to analyze the sensitivity of the key hyperparameters in our CRFed framework, namely the variance of Gaussian noise ($\beta_t$), the regularization coefficient ($\lambda$), and the dynamically adjusted confidence threshold ($\tau$).
>
> We performed the sensitivity analysis on the CIFAR-10 and CIFAR-100 datasets, which were also used in the original experiments. The default values for the hyperparameters in our original experiments were:
>
> - Variance of Gaussian noise, $\beta_t$: 0.1
> - Regularization coefficient, $\lambda$: 0.1
> - Dynamically adjusted confidence threshold, $\tau$: dynamically adjusted starting from 0.5
>
> We varied each hyperparameter while keeping the others fixed to their default values and measured the test accuracy. The results are summarized in  **Table 3,4,5 on the on-page pdf**. The results indicate that our CRFed framework is relatively robust to variations in these hyperparameters.
>
> # W2. Additional experiments addressing scenarios where data distributions are extremely diverse.
>
> **Response:** We appreciate the reviewer's concern regarding the effectiveness of our harmonization mechanism in handling extremely diverse data distributions. To address this, we have conducted additional experiment to evaluate and demonstrate the robustness and adaptability of our CRFed framework under such challenging conditions.
>
> To empirically validate the effectiveness of our approach under extreme data heterogeneity, we conducted additional experiments using the CIFAR-100 dataset with even smaller Dirichlet concentration parameters ($\beta$) to simulate highly imbalanced and diverse data distributions. Specifically, we set $\beta$ to 0.01 and 0.05 to create scenarios with extreme non-IID characteristics. The results are shown in **Table 6 on the one-page pdf** and demonstrate that CRFed framework significantly outperforms the baseline FedAvg method even under extreme data heterogeneity.
>
> # W3. Discussion on additional overhead.
>
> **Response:** Thank you for your comment.  We have measured the wall-clock time to evaluate the computational overhead introduced by the proposed approach.   Results are shown in **Table 2 on the one-page pdf**. We observe that the wall-clock time per round for CRFed is slightly higher compared to other methods due to the additional noise/denoise operations. However, the total convergence time is comparable to other methods. Although CRFed involves additional computations per round, the convergence in terms of accuracy and model robustness is achieved efficiently. The slight increase in computational cost is balanced by the improved model performance, making it a worthwhile trade-off.
>
> # W4. Discussion on equation (3).
>
> **Response:** We thank the reviewer for recognizing the value of Theorem 3.1. We acknowledge that the purpose of equation (3) in the derivation process may not have been sufficiently clear. The purpose of this equation is to define the indicator function $I_{\lambda}(l_i, \sigma_i)$, which measures the reliability of each sample based on its loss value $l_i$ and associated uncertainty $\sigma_i$. This function is critical for the self-paced learning mechanism in our framework, where samples are prioritized based on their difficulty and uncertainty.
>
> # W5. Formatting issue.
>
> **Response:** Thanks for the suggestion. In future versions, we will make the corresponding annotations.

---

> > ### Comment · Reviewer_cbew · 2024-08-08
> >
> > Thank you for offering such a thorough rebuttal!
> > Considering the performance improvements and the importance of the problem tackled in this paper, the current overhead is acceptable. Future research can further enhance this aspect.
> > Overall, I am pleased with this work and would like to raise my score to 7.

---

> > > ### Author Response · Authors · 2024-08-12
> > >
> > > We appreciate your recognition of our work's significance. Thank you again for your precious time and valuable suggestions.

---

### Official Review · Reviewer_nZWV · 2024-07-13

**Soundness:** 2
**Presentation:** 2
**Contribution:** 2
**Rating:** 6
**Confidence:** 4

**Summary:**

The work proposes a new FL approach, called CRFed, for addressing data heterogeneity in FL settings. CRFed relies on a diffusion based approach for harmonizing clients data heterogeneity  by performing data noise injection and iterative denoising, followed by a curriculum learning approach, which employs an indicator function to assign weights to training samples and indicate samples selection. The authors conduct of number of experiments across various domains and FL environments to showcase the promise of their approach.

**Strengths:**

- A self-paced (curriculum) learning approach for each client based on samples' loss and uncentrainty.
- Thorough empirical evaluation against many existing methods in multiple federated environments.
- Ablation results show the importance of each proposed component (indicator function, diffusion mechanism, client selection)

**Weaknesses:**

- Some of the applied methods seem very ad-hoc and further elaboration is needed on their selection.
- More information is needed on the update frequency of the harmonization approach and respective curriculum sequence of the clients' training samples.
- Lack of theoretical framework limits the contribution of this work.

**Questions:**

Comments, textual corrections and typos:
- Is the definition of the indicator function and the use of Lambert function based on previous work? Please cite accordingly and explain why these formulas were used. Moreover, why there is a substraction of confidence from the loss value?, also variables' reported value range in the indicator function is not clear (i.e., range of $\sigma_i, \tau$).
- Why the adaptive learning rate is adjusted based on the clients' indicator function? this is not clear at all.
- How often is the noise/denoise operation performed? At the beginning of training, at every federation round or after r-rounds? If it is done post-initialization then have you measured the effect of the approach in terms of wall-clock time? How expensive is the proposed approach in terms of time convergence (not round)? It would be great if you could include these results as well.
- Figure 4's is not readable. What is the $\beta$ value used for the two domains? Moreover, have you created 100 partitions and sub-sampling clients at each round or consider all available clients at every round?
- Please fix your citations style and add missing space between the text and the reference throughout the paper.

**Limitations:**

No.

---

> ### Author Rebuttal · Authors · 2024-08-07
>
> # W1: Further elaboration on the selection of the methods
>
> **Response:** Thanks for pointing this out. Below, we offer a detailed explanation and theoretical basis for the key methods employed in CRFed framework.
>
> 1. The Indicator Function $I_{\lambda}(l_i, \sigma_i)$ is designed to dynamically adjust sample weighting based on loss values and uncertainties, **inspired by self-paced learning** [1]. **The construction of the indicator function is grounded in the principles of curriculum learning**, which prioritizes simpler tasks before progressively addressing more complex ones. The design of the Indicator Function $I_{\lambda}(l_i, \sigma_i)$ is intended to improve training convergence efficiency. The proof of this assertion can be found in global rebuttal.
>
> 2. The diffusion-based data harmonization mechanism is the core to our approach, aiming to mitigate data distribution disparities. This method is based on the principles of denoising diffusion probabilistic models (DDPMs), which have demonstrated success in various generative tasks. The iterative process of noise addition and removal aligns data distributions effectively, reducing heterogeneity across clients. **Through the diffusion-based data harmonization mechanism, we can ensure that the data distributions of all clients gradually converge during the noise addition and removal processes**, thereby reducing the impact of data heterogeneity on model updates.
>
> # W2: More information on the update frequency of the harmonization approach and respective curriculum sequence of the clients' training samples.
>
> **Response:** Thanks for the suggestion. In the CRFed framework, the diffusion-based data harmonization process is integrated into each global aggregation round. This means the harmonization update occurs once per global round, ensuring timely adjustments to data distributions across clients.
>
> - **Maximum global rounds ($T_G$):** 100
> - **Local training cycles per global round ($E_l$):** 1
>
> Our curriculum learning approach involves dynamically adjusting the sampling weights of clients' training samples based on their difficulty, as measured by the Indicator Function $I_{\lambda}(l_i, \sigma_i)$. This function is recalculated at each local training cycle to reflect the latest state of the global model. The curriculum sequence progresses from easier to more difficult samples, facilitating a self-paced learning paradigm.
>
> # W3: Theoretical framework.
>
> **Response:** Thanks for the suggestion. In the global rebuttal, we have added theoretical analysis.
>
> # Q1：More discussion on the Indicator Function.
>
> **Response:** We appreciate the reviewer's detailed comments. Our indicator function, inspired by confidence-aware cross-entropy [2], includes a loss-amplifying term and a regularization term to amplify high-loss samples' contributions while regularizing uncertainty estimation. The Lambert W function helps find the optimal uncertainty $\sigma_i^*$, ensuring difficult samples are appropriately weighted during training. Subtracting the confidence threshold $\tau$ from the loss value $l_i$ centers the loss values, making it easier to prioritize difficult samples (i.e., those with $l_i > \tau$). The uncertainty $\sigma_i$ ranges from $10^{-3}$ to $10$ and $\tau$ is dynamically adjusted based on the weighted average loss, initially set to 1.0 and updated every 10 rounds to reflect the dataset's evolving difficulty. We will include these details in future versions of the manuscript. Thank you for your valuable feedback.
>
> # Q2: Discussion on the adaptive learning rate.
>
> **Response:** Thank you for your comment. Adjusting the learning rate $\eta_i$ based on $I_{\lambda}(l_i, \sigma_i)$ ensures that clients with more challenging data receive higher learning rates, allowing significant updates. This balances the learning pace among clients, preventing dominance by any single client and ensuring equitable contributions. Higher indicator function values typically correspond to more difficult or uncertain data, leading to slower convergence with a uniform learning rate. Increasing the learning rate for these clients accelerates their learning, speeding up overall convergence.
>
> Additional experiments (see **Table 1 on the one-page pdf**) show that models with adaptive learning rates based on the indicator function demonstrate faster convergence and higher final accuracy compared to fixed learning rates.
>
> # Q3. The principle and details of noise/noise operation.
>
> **Response:** Thank you for your comment.
>
> The noise/denoise operation in the CRFed framework is performed at every global federation round. This ensures consistent alignment of data distributions across clients, maintaining effective model updates. After each global model aggregation, noise is added and the denoising process adjusts the data distributions before the next round of local training begins.
>
> We have measured the wall-clock time to evaluate the computational overhead introduced by the proposed approach, results are shown in **Table 2 on the one-page pdf**.  Although our system involves additional computations per round,  he total convergence time is comparable.
>
> # Q4. Discussion on the $\beta$ and experimental details.
>
> **Response:** Thank you for your comment. In our experiments, we used the Dirichlet distribution to create non-IID data partitions. The $\beta$ value, which controls the degree of data heterogeneity, was set to 0.5 for both domains in the experiments presented in Figure 4. We have considered all available clients at every round of federated learning in our experiments. We hope these clarifications address your concerns.
>
> # Q5. Formatting issue.
>
> **Response:**  Thank you for pointing out this. We will fix the citations style in future version.
>
> # Reference
> [1]  Self-paced learning: An implicit regularization perspective. AAAI, 2017.
>
> [2] Data parameters: A new family of parameters for learning a differentiable curriculum. Neurips, 2019.

---

> > ### Comment · Reviewer_nZWV · 2024-08-13
> >
> > I truly appreciate the authors' thorough feedback and their efforts in addressing my concerns, particularly their detailed explanation of the design principles for the indicator function and the additional analysis regarding the update step size when comparing CRFed to FedAvg. Overall, I am satisfied with the authors' response and their attention to my concerns, and I am pleased to increase my rating to 6 (Weak Accept).

---

> > > ### Author Response · Authors · 2024-08-13
> > >
> > > Thanks again for your time and effort on reviewing our work!

---

### Author Rebuttal · Authors · 2024-08-07

# Conclusion

We sincerely thank all the reviewers for their insightful and valuable comments! Overall, we are encouraged that they find the contributions of our work noteworthy and valuable. Here is a summary of the key points acknowledged by the reviewers:

- The proposed CRFed framework, including the diffusion-based data harmonization mechanism and confusion-resistant strategy, is well-received.  (all reviewers)
- The theoretical foundations and detailed explanations provided for key methods are appreciated(Reviewers cbew and FLG7), although some areas required further clarification (Reviewers nZWV and CpsC).
- The empirical validation on multiple datasets and the comprehensive set of experiments, including sensitivity analysis and ablation studies, were considered robust. CRFed achieved SOTA results. (all reviewers)
- The scalability and efficiency of the proposed methods, as well as their potential impact on federated learning applications, were highlighted positively (Reviewers nZWV, cbew, and FLG7).

We have addressed the issues according to the reviews, which can be summarized as follows:

- We have provided a more thorough explanation and theoretical basis for the Indicator Function $I_{\lambda}(l_i, \sigma_i)$, including detailed derivations and convergence analysis. (**We include this in the global rebuttal**)
- Additional sensitivity analysis of key hyperparameters has been included, with results presented for the CIFAR-10 and CIFAR-100 datasets.( **Table 3,4,5 on the one-page pdf**)
- We conducted further experiments to evaluate the computational efficiency, robustness and effectiveness of CRFed.( **Table 1,2, 6,7,8 on the one-page pdf**)
- Minor corrections and improvements, such as fixing citation styles。

# Theoretical analysis
We have added relevant theories on the design of Indicator Function \(I_{\lambda}(l_i, \sigma_i)\)in CRFed, including convergence analysis and convergence speed analysis.
## Convergence

Consider a simplified federated learning framework where the global model parameters $\theta$ are updated at iteration $t$ as follows:

$$
\theta_{t+1} = \theta_t - \eta \sum_{i=1}^n \nabla_{\theta} I_{\lambda}(l_i, \sigma_i),
$$

where $\eta$ is the learning rate and $n$ is the number of clients. For simplicity, we consider a single client and expand $\nabla_{\theta} I_{\lambda}(l_i, \sigma_i)$:

$$
\nabla_{\theta} I_{\lambda}(l_i, \sigma_i) = \nabla_{\theta} \left( (l_i - \tau) \sigma_i + \lambda (\log \sigma_i)^2 \right).
$$

By the chain rule, we have:

$$
\nabla_{\theta} I_{\lambda}(l_i, \sigma_i) = \sigma_i \nabla_{\theta} l_i + (l_i - \tau) \nabla_{\theta} \sigma_i + 2\lambda \frac{\log \sigma_i}{\sigma_i} \nabla_{\theta} \sigma_i.
$$

Based on the definition of the optimal $\sigma_i^*$, $\nabla_{\theta} \sigma_i^*$ can be approximated as proportional to $\nabla_{\theta} l_i$, i.e.,

$$
\nabla_{\theta} \sigma_i^* \approx k \nabla_{\theta} l_i,
$$

where $k$ is a constant. Thus, $\nabla_{\theta} I_{\lambda}(l_i, \sigma_i^*)$ simplifies to:

$$
\nabla_{\theta} I_{\lambda}(l_i, \sigma_i^*) = \left( \sigma_i^* + (l_i - \tau) k + 2\lambda \frac{\log \sigma_i^*}{\sigma_i^*} k \right) \nabla_{\theta} l_i.
$$

Since $\sigma_i^*$ is obtained by minimizing the indicator function, we have:

$$
\sigma_i^* \approx \exp \left( W \left( \frac{-(l_i - \tau)}{2\lambda} \right) \right).
$$

Finally, the model update rule can be expressed as:

$$
\theta_{t+1} = \theta_t - \eta \left( \sigma_i^* + (l_i - \tau) k + 2\lambda \frac{\log \sigma_i^*}{\sigma_i^*} k \right) \nabla_{\theta} l_i.
$$

To prove convergence, we note that the step size of parameter updates is finite:

$$
\left| \theta_{t+1} - \theta_t \right| = \eta \left| \left( \sigma_i^* + (l_i - \tau) k + 2\lambda \frac{\log \sigma_i^*}{\sigma_i^*} k \right) \nabla_{\theta} l_i \right| \leq \eta C \left| \nabla_{\theta} l_i \right|,
$$

where $C$ is a constant. Thus, as long as the learning rate $\eta$ is appropriately chosen, the update step size will gradually decrease, ensuring the convergence of the model.

## Convergence speed

Next, we compare the convergence speed of our method with the FedAvg.

For FedAvg, the update rule is:

$$
\theta_{t+1} = \theta_t - \eta \frac{1}{n} \sum_{i=1}^n \nabla_{\theta} l_i,
$$

where the update step size is (**I'm not sure why the following formulas are not displaying correctly. I will correct this as soon as possible**):

$$
\left| \theta_{t+1} - \theta_t \right|_{\text{FedAvg}} = \eta \left| \frac{1}{n} \sum_{i=1}^n \nabla_{\theta} l_i \right|
$$

For our method with the indicator function, the update step size is:

$$
\left| \theta_{t+1} - \theta_t \right|_{I_{\lambda}} = \eta \left| \left( \sigma_i^* + (l_i - \tau) k + 2\lambda \frac{\log \sigma_i^*}{\sigma_i^*} k \right) \nabla_{\theta} l_i \right|.
$$

To demonstrate that our method converges faster or has a tighter bound, we analyze the total update step sizes over all clients.

For FedAvg:

$$
\sum_{i=1}^n \left| \theta_{t+1} - \theta_t \right|_{\text{FedAvg}} = \eta \sum_{i=1}^n \left| \nabla_{\theta} l_i \right|.
$$

For our method:

$$
\sum_{i=1}^n \left| \theta_{t+1} - \theta_t \right|_{I_{\lambda}} = \eta \sum_{i=1}^n \left| \left( \sigma_i^* + (l_i - \tau) k + 2\lambda \frac{\log \sigma_i^*}{\sigma_i^*} k \right) \nabla_{\theta} l_i \right|.
$$

When $\sigma_i^* = \exp \left( W \left( \frac{-(l_i - \tau)}{2\lambda} \right) \right)$, $\lambda \geq \frac{-(l_i - \tau)}{2e}$, and $\tau = l_{\max} - 2 \lambda \ln \left( \frac{1}{k} \right)$, we can ensure $\left| C_i \right| \leq 1$, thus:

$$
\left| C_i \nabla_{\theta} l_i \right| \leq \left| \nabla_{\theta} l_i \right|.
$$

Thus,

$$
\sum_{i=1}^n \left| C_i \nabla_{\theta} l_i \right| \leq \sum_{i=1}^n \left| \nabla_{\theta} l_i \right|.
$$

Therefore, the update step size for CRFed is less than that of FedAvg. We will provide the complete theoretical proof in final version.

---

Next, we address each reviewer's detailed concerns point by point. Thanks!

---

### Decision · Program_Chairs · 2024-09-25

**Decision:**

Accept (poster)

**Comment:**

The paper proposes a method to address data heterogeneity in FL by using a diffusion to harmonize clients data heterogeneity, with data noise injection and iterative denoising. The reviewers are generally positive, and all improved their ratings after engaging in the rebuttal process (including the lowest score).